# Coplanar Electrode Polymer Modulators Incorporating Fluorinated Polyimide Backbone Electro-Optic Polymer

**Eun-Su Lee [1], Sung-Moon Kim [1], Mi-Hye Yi [2], Jae-Won Ka [2] and Min-Cheol Oh [1,*]** 

[1] Department of Electronics Engineering, Pusan National University, Pusan 46241, Korea;
    ensulee@pusan.ac.kr (E.-S.L.); sungmoon@pusan.ac.kr (S.-M.K.)
[2] Advanced Functional Polymers Research Center, Korea Research Institute of Chemical Technology,
    Daejeon 34114, Korea; mhyi@krict.re.kr (M.-H.Y.); jwka@krict.re.kr (J.-W.K.)
[*] Correspondence: mincheoloh@pusan.ac.kr; Tel.: +82-51-510-2358

**Abstract:** High-speed coherent optical communication has been expanding to handle the ever-increasing data traffic, and the large modulation bandwidth of electro-optic (EO) polymer modulators has been especially appreciated. However, to be useful in optical communication, the EO polymer device should address several issues, such as thermal stability, photo-oxidation, and bias drift. In this work, as a part of the experiments to address these challenges, an EO polymer with a fluorinated polyimide backbone is utilized to create EO polymer modulators with improved thermal stability. A coplanar electrode structure is introduced to enhance the poling efficiency and reduce the bias drift.

**Keywords:** electro-optic polymer; polymer waveguide devices; integrated optics

## 1. Introduction

Electro-optic (EO) polymer modulators have an original advantage of small velocity mismatch between radio-frequency (RF) and optical signals, and by employing a traveling-wave modulation electrode, an optical modulator operable above 100 Gbps was demonstrated [1–3]. Although research on EO polymers has been conducted for decades, to prove the technology to be useful for practical application, there are issues to be addressed, such as photo-oxidation, bias drift, large insertion loss, and poor thermal stability [4–6].

The photo-oxidation problem can be relaxed by using commercial hermetic packaging, which has been widely adopted for commercial optical devices with much reduced cost recently. The bias drift is mainly caused by the field-assisted drift of impurity ions remaining in the EO polymer material during the material synthesis. The impurity ions can be reduced significantly by employing highly advanced impurity-filtering processes during the material preparation step. Furthermore, the bias-point drift is no longer a concern because recent coherent communication replaces the intensity modulators with phase modulators that do not require bias control. The insertion loss due to the material absorption can be reduced by using fluorinated polymers, which reduces the vibrational absorption for the optical communication wavelengths. The thermal stability can be improved by adopting ridged-backbone polymers such as polyimides.

Various studies have been conducted to improve the EO coefficient and thermal stability. Several approaches aimed to increase the glass transition temperature ($T_g$) of poly(methyl methacrylate) (PMMA)-based EO polymers and improve the thermal stability of the device, in which the EO coefficient ($r_{33}$) was over 100 pm/V and preserved at 105 °C for 2000 h [7]. A high EO coefficient was

achieved by using novel chromophores, showing a maximum $r_{33}$ of 290 pm/V, and maintaining 99% of the initial $r_{33}$ at 85 °C for 500 h [8]. High poling and modulation efficiencies were achieved by taking advantage of the highly confined optical mode in the Si slot waveguide [9]. State-of-the-art Si-slot EO modulators demonstrated 200 Gbps four-level pulse amplitude modulation (PAM-4) modulation with a low voltage–length product ($V_\pi \cdot L$) of 0.41 V·mm and total insertion loss of 0.7 dB [10].

Several studies were conducted using polyimides in EO modulators. An optical modulator with propagation loss of 1.14 dB/cm and $V_\pi$ of 5.4 V was reported, which employed a guest–host-type polymer [11,12]. The chromophores in the guest–host EO polymer could be aligned relatively easily, even in low poling fields, and thus, the EO modulator could achieve low $V_\pi$. However, it was difficult to maintain the poled order of the chromophores at elevated temperatures. Side-chain EO polymers were confirmed to exhibit superior thermal stability compared to guest–host EO polymers [1,7,13]. Another study was conducted with a polyimide backbone sidechain EO polymer; however, the fabricated modulator exhibited high losses because the polyimide backbone exhibited large absorption [14].

Fluorinated polymers can reduce the loss of optical signals, caused by the C–H vibrational overtone at a wavelength of 1550 nm [11,15–17]. Therefore, fluorinated polyimides have been widely used as a substrate for flexible organic light-emitting diode (OLED) devices owing to their high transparency, good thermal stability, and excellent mechanical and chemical properties [17]. The EO polymer used in this study is based on a fluorinated polyimide synthesized from 2,2-Bis (3-amino-4-hydroxyphenyl) hexafluoropropane (6FAP) and 4,4′-(hexafluoroisopropylidene)-diphthalic anhydride (6FDA) [18].

In this work, we demonstrate EO polymer devices by incorporating a fluorinated polyimide sidechain EO polymer. Polarization modulators and Mach–Zehnder (MZ) intensity modulators are fabricated to measure the EO coefficient of the device and analyze the poling efficiency. We confirm that the coplanar electrode can effectively reduce the bias drift and increase the poling efficiency by discarding the voltage drop across the cladding layers.

## 2. Preparation of Fluorinated Polyimide EO Polymer and Design of Waveguide Devices with Coplanar Electrodes

The sidechain EO polymer material with fluorinated polyimide backbone, depicted in Figure 1, was supplied from Korea Research Institute of Chemical Technology, Daejeon, South Korea [18]. A hydroxy polyimide was synthesized by the polymerization of the reactants at 150 °C for 3 h. Commercially available DR1 chromophores were selected to obtain the EO property because the chromophores could not be synthesized. The synthesized polymers were post-functionalized by covalently bonding the DR1 chromophores, to yield the EO polymer. The material was supplied in the form of dry red powder and was dissolved in cyclohexanone to produce a 15 wt% solution. The solution was spin-coated on a Si wafer at 700 rpm for 20 s, and baked on a hotplate for 10 min at 80 °C, and for 30 min at 200 °C. The EO coefficient of the thin film, measured by using the reflection method, was 27.7 pm/V at 1310 nm, and the $T_g$ was exceptionally high as 202 °C. Thermal tests were conducted to confirm that over 90% of the initial $r_{33}$ is maintained even after heating at 150 °C for 500 h, proving the superior thermal stability of the material.

The final thickness of the baked film was 3.7 μm. The film presented excellent surface morphology with a low surface roughness less than 5 nm. The surface roughness of the film was measured using an atomic force microscope, as shown in Figure 2a. The average surface roughness ($R_a$) was measured as 557 pm, as shown in Figure 2b. The refractive index of the EO polymer film was measured by the prism coupling method with a gadolinium gallium garnet (GGG) prism as 1.6252 and 1.6198 for TE and TM, respectively, for the wavelength of 1550 nm.

The refractive indices of the materials selected for the waveguide device were as follows: 1.445 for $SiO_2$, the lower cladding material; 1.625 for the EO polymer; and 1.501 for NOA74, the upper cladding material. According to the effective index method, rib-structured waveguides with waveguide widths of 4 and 6 μm satisfied the single-mode condition.

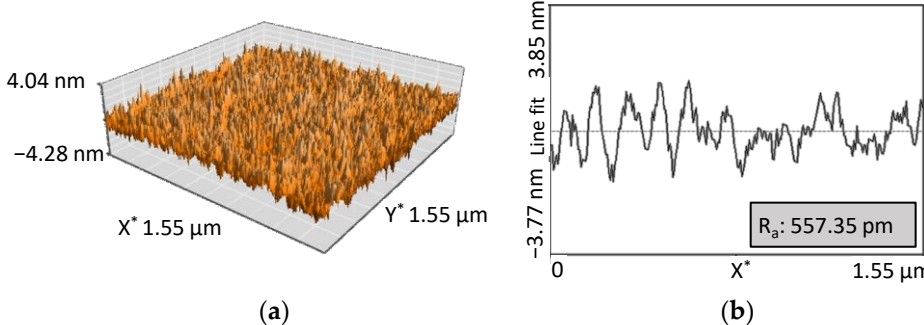

**Figure 1.** Scheme of the electro-optic (EO) polymer molecular structure used in this work, showing the fluorinated polyimide (black) and the DR1 chromophore side chain covalently bonded to the polyimide (red).

(**a**)

(**b**)

**Figure 2.** (**a**) Surface roughness of EO polymer film measured using an atomic force microscope. (**b**) Cross-sectional plot of the surface roughness data.

For a straight waveguide, polarization modulators could be produced by placing coplanar electrodes as shown in Figure 3a. The gap between the electrodes and waveguide had to be close enough to increase the poling and modulation efficiencies until the surface plasmon absorption was not arising. For an input polarization of 45°, both TE and TM modes were excited, and the output polarization depended on the phase retardation between the two polarizations owing to the difference in the EO coefficients of $r_{33}$ and $r_{13}$. MZ interferometers were designed as shown in Figure 3b, by placing the coplanar electrodes along the two straight waveguides. The modulation efficiency of the MZ modulator could be doubled by a push–pull operation, where the two waveguides were poled in same direction and operated in opposite directions, as depicted in Figure 3c.

In the case of vertical poling devices, the electric field crosses the core and cladding materials. A core material doped with chromophores generally exhibits lower resistivity than the cladding material, at an elevated temperature, causing large voltage drops across the cladding layer; thus, the poling field in the core layer would decrease at a high temperature. In coplanar electrode devices, the poling field across the core layer is not affected by the difference in the material resistivity. The poling field distribution of coplanar poling was calculated by the finite element method, assuming that the dielectric constants were uniform throughout the film. The electric potential and field distribution for the applied poling field were calculated as shown in Figure 3d. The middle electrode was left floating. The poling field at the cores of the two waveguides was parallel to the coplanar electrodes. At vertical positions of the waveguide center and electrode, the electric field intensities along horizontal direction were found as shown in Figure 3e, where the applied voltage ($V_0$) was 1 kV and the distance between the edges of the two electrodes ($d_g$) was 10 μm. The maximum electric field was produced near the edge of the electrode, and that its magnitude decreased further away from the electrode. The magnitude of the effective electric field at the center of the core became 36 V/μm, which was lower than the value expected by uniform field assumption, 50 V/μm (1 kV/20 μm).

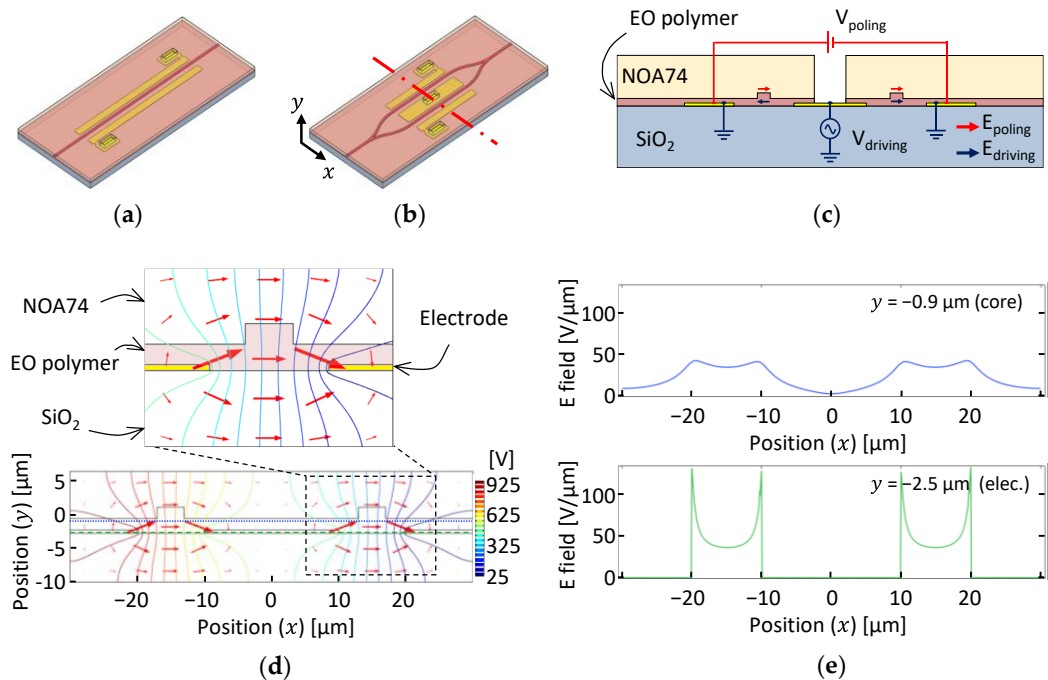

**Figure 3.** Overview of a (**a**) straight waveguide polarization modulator device and (**b**) Mach–Zehnder intensity modulator device. (**c**) Device cross-section showing the directions of the poling field and driving field for push–pull operation. (**d**) Electric potential and field distributions produced by the coplanar electrodes. (**e**) Electric field plot along the horizontal direction at the vertical positions of the waveguide center (along the dotted line in (**d**)) and the electrode (along the dashed line in (**d**)).

## 3. Fabrication of The EO Polymer Modulator

Prior to the device fabrication, multilayer film coatings were evaluated by coating several materials on top of the annealed EO polymer film, to determine the process compatibility. When a photoresist was coated over the EO polymer film, severe cracking occurred, because of the solvent attack. When low-loss fluorinated polymer ZPU produced by ChemOptics Inc., Daejeon, South Korea, was used as the upper cladding material, cracks appeared after the UV curing of the ZPU polymer film. Finally, UV-curable epoxy NOA74 from Norland Optical Adhesives was selected as the upper cladding material because no damage was found on the EO polymer film even after the UV curing of NOA74.

The fabrication process is summarized in Figure 4. Si substrate with a 14.5 μm oxide layer grown by wet thermal oxidation is used as both the optical cladding and electrical insulating layer. Cr–Au layers of 10–60 nm are deposited to form the electrode. The lift-off process is adopted for electrode fabrication, to prevent the chance of electrical shorts due to the metal residue. The EO polymer is coated on top of the electrode layer producing a 3.7 μm film thickness. Then, NOA74 as a protection layer is covered on the EO polymer with a thickness of 1.0 μm. After the photolithography to define the waveguide pattern, the core layer is etched by 1.2 μm in an oxygen plasma to form a rib waveguide. Then, the upper cladding of 5 μm is formed using NOA74. The metal pads for poling and operation are opened by dry etching the polymer layers through a fine metal mask.

To reduce the burden of experimental work in optimizing the poling process, we designed the electrodes as shown in Figure 5a. Three pairs of electrodes with different gaps (12, 16, and 20 μm) were connected in parallel to form the same potential difference across them and perform poling simultaneously. When the poling voltage was increased slowly, the device with a narrow electrode gap experienced the highest electric field first and proceeded to breakdown if the voltage was increased beyond a certain extent. Then, we presume that a maximum field before the breakdown was applied to the devices with wider electrode gaps.

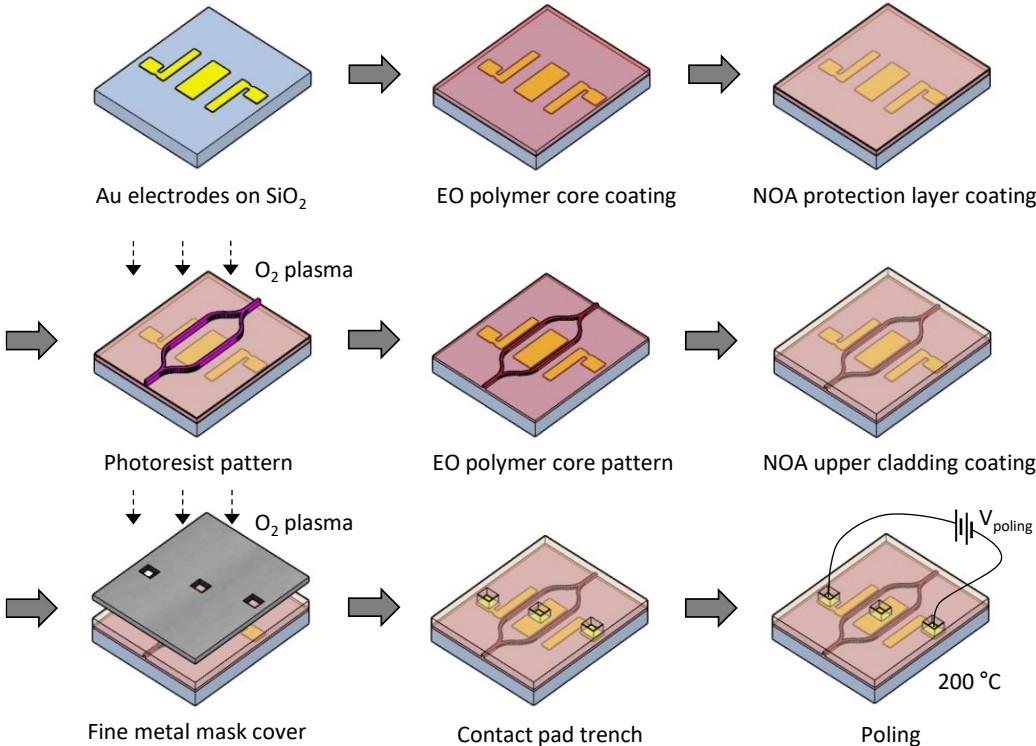

**Figure 4.** Schematic fabrication procedure of the EO polymer Mach–Zehnder modulator with a coplanar electrode.

In the poling setup shown in Figure 5b, spring contact probes were used. The pads were covered lightly with silver paste, to ensure contact without damaging the thin metal pads. The sample was poled on a hot plate inside a nitrogen purge chamber to prevent the oxidation of the chromophore molecules at high temperatures and remove the effects of moisture [4]. The oxygen concentration inside the chamber reached 1.6% after 10 min of $N_2$ purging. The amount of current flowing through the EO polymer during the poling process was measured using a picoammeter (Keithley 6485). A resistor was placed as a current-limiting load after the DUT and two diodes in opposite directions were connected in parallel with the picoammeter to prevent accidental high currents through the picoammeter. As the temperature increased, the resistance of the EO polymer dropped, causing an increase in the current flow.

In the first few trials of poling, the voltage was increased to a high level first, prior to the temperature elevation, as shown in Figure 5c. In this case, an early breakdown was observed on the electrode, as shown in Figure 5d, caused by a defect near the electrode. Electro-optic properties were barely observed in this device. In the second case, the temperature was increased first, then the voltage was raised, as shown in Figure 5e. An excessive amount of current suddenly flowed because of the rapid voltage increase at the poling temperature, which led to dielectric breakdown, as shown in Figure 5f.

The poling process was optimized to limit the maximum current flow during the poling, by controlling the temperature and voltage in several steps, as shown in Figure 5g. At the first step, the voltage was increased gradually up to 1 kV, which was guaranteed as a safe voltage from the previous experiments. By observing the current flow at 200 °C, the voltage was gradually increased to the maximum poling voltage of 1.8 kV, corresponding to an electric field of 100 V/µm. Thus, we were able to avoid breakdown and achieve the highest poling efficiency.

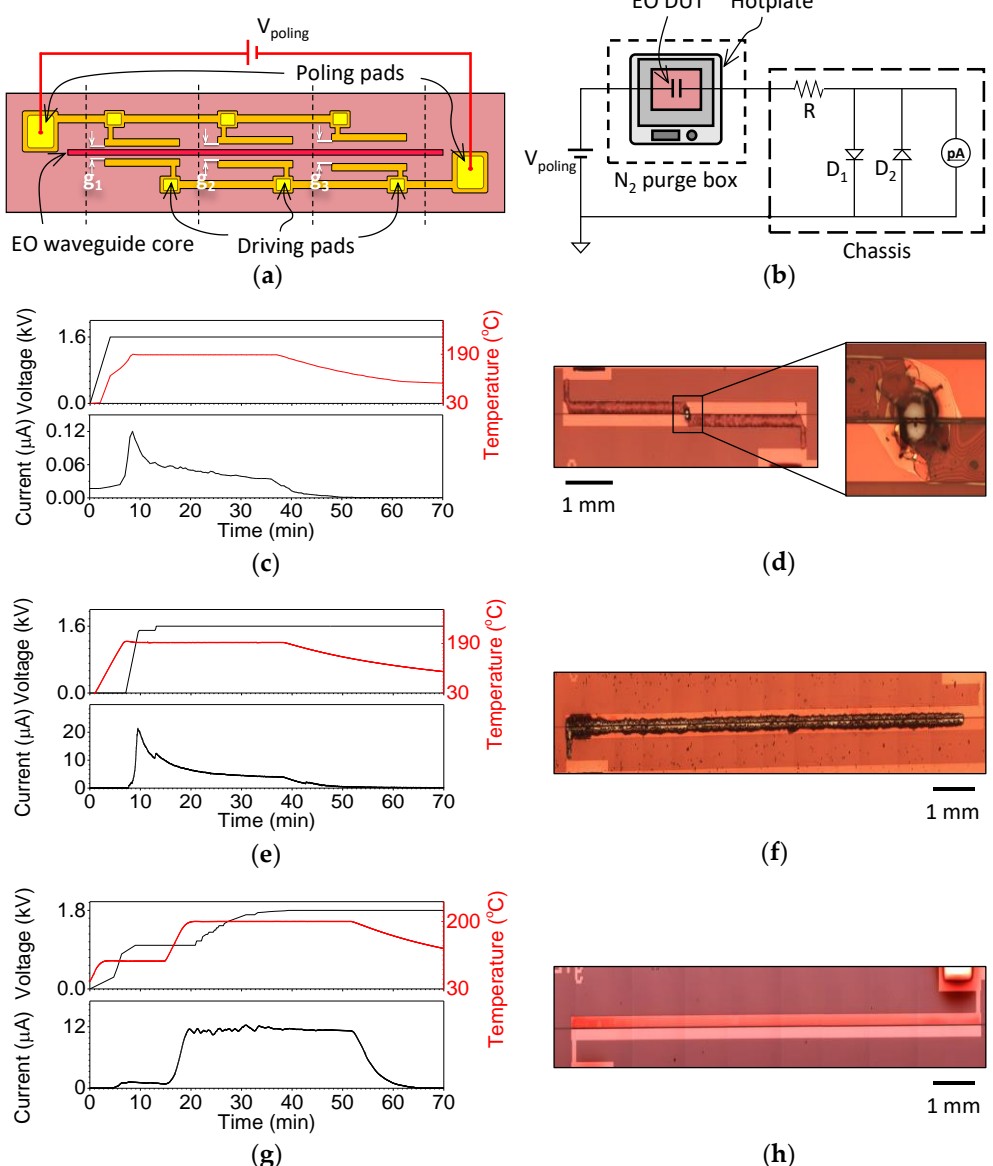

**Figure 5.** (**a**) Electrode design for efficient poling experiment, where three coplanar electrodes with different gaps are connected in parallel. Note that the aspect ratio of the electrode gap is exaggerated to show the concept clearly. (**b**) In the poling setup, the device is placed on a hotplate inside a nitrogen purge box connected to a picoammeter with an accidental high-current protection circuit. The voltage, temperature, and current profiles during the poling process and microscopic images of each sample: (**c**) profile when the voltage is increased to a high level before increasing the temperature; (**d**) electrode destroyed because of a defect near the electrode; (**e**) rapid rise in the voltage to a high poling voltage causing breakdown; (**f**) dielectric breakdown along the electrode; (**g**) stable poling profile without breakdown, by stepwise voltage and temperature control; (**h**) sample that survived the poling process.

## 4. Measurement of the Optical Waveguide and MZ Modulator Performance

The insertion losses of the devices with different lengths were measured using a 1550 nm distributed feedback (DFB) laser, as shown in Figure 6. Single-mode fibers were aligned with the device, and a fiber-optic polarization controller was used to adjust the input polarization to TE or TM. The propagation losses of a 4-μm-core waveguide were measured as 1.9 and 2.2 dB/cm for TE and TM, respectively. For a 6-μm-core waveguide, the propagation losses measured for TE and TM polarizations were 1.4 and 1.6 dB/cm, respectively, which indicated that the 6-μm waveguide had better mode

confinement. The coupling losses of the 4- and 6-µm waveguides were 3.3 and 2.9 dB/facet, respectively. From comparisons with the measurement results of the unpoled samples, the poling-induced loss was approximately 0.5 dB/cm.

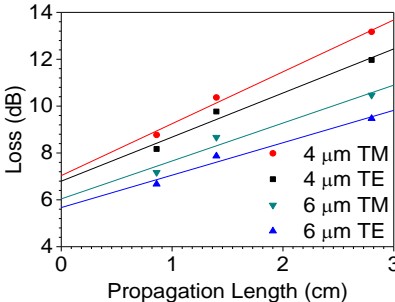

**Figure 6.** Insertion loss measurement results using cut-back method.

To observe the polarization modulation, caused by the difference between $r_{13}$ and $r_{33}$ EO coefficients, a straight waveguide modulator was characterized with a 1550 nm DFB source, as shown in Figure 7a. The length of device was 10 mm with an electrode length of 5 mm. The input polarization was adjusted to +45° linear polarization, and the output polarization was observed using a CCD through a −45° free-space polarizer to confirm the intensity modulation due to the polarization change. Then, the output polarization was monitored using a polarization analyzer (General Photonics, POD-101), and $2\pi$ phase retardation was observed between the TE and TM modes, as shown in Figure 8b. Assuming $r_{33} \approx 3 \, r_{13}$, $r_{33}$ could be calculated from the measured $V_\pi$ for complete polarization conversion [19]. For a poling voltage of 1.8 kV, the actual poling field at the center of the waveguide core became 100, 77, and 63 V/µm, corresponding to electrode gaps of 12, 16, and 20 µm, respectively. The 12 µm gap electrode showed a $V_\pi$ of 128 V, and the obtained $r_{33}$ value was 8.42 pm/V, as shown in Figure 7c.

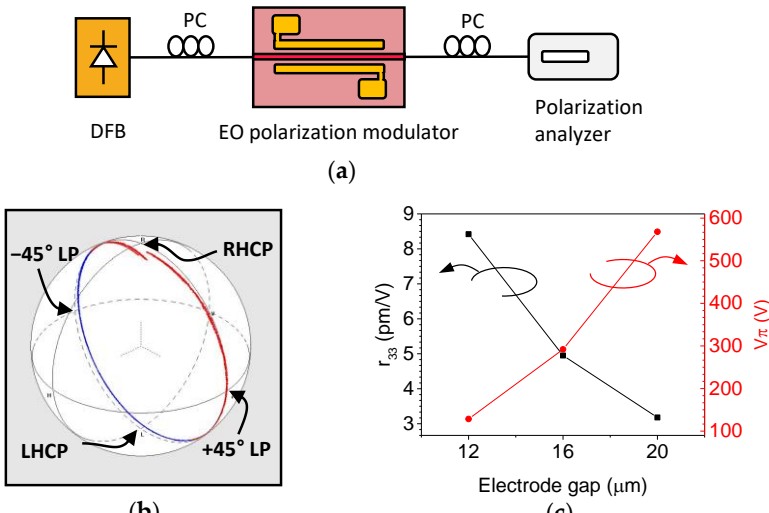

**Figure 7.** Polarization conversion efficiency measurement using a straight waveguide polarization modulator: (**a**) experimental setup; (**b**) output polarization states measured during polarization modulation, drawn on the Poincare sphere; and (**c**) measured $V_\pi$ and calculated $r_{33}$ of poled samples with different electrode gaps.

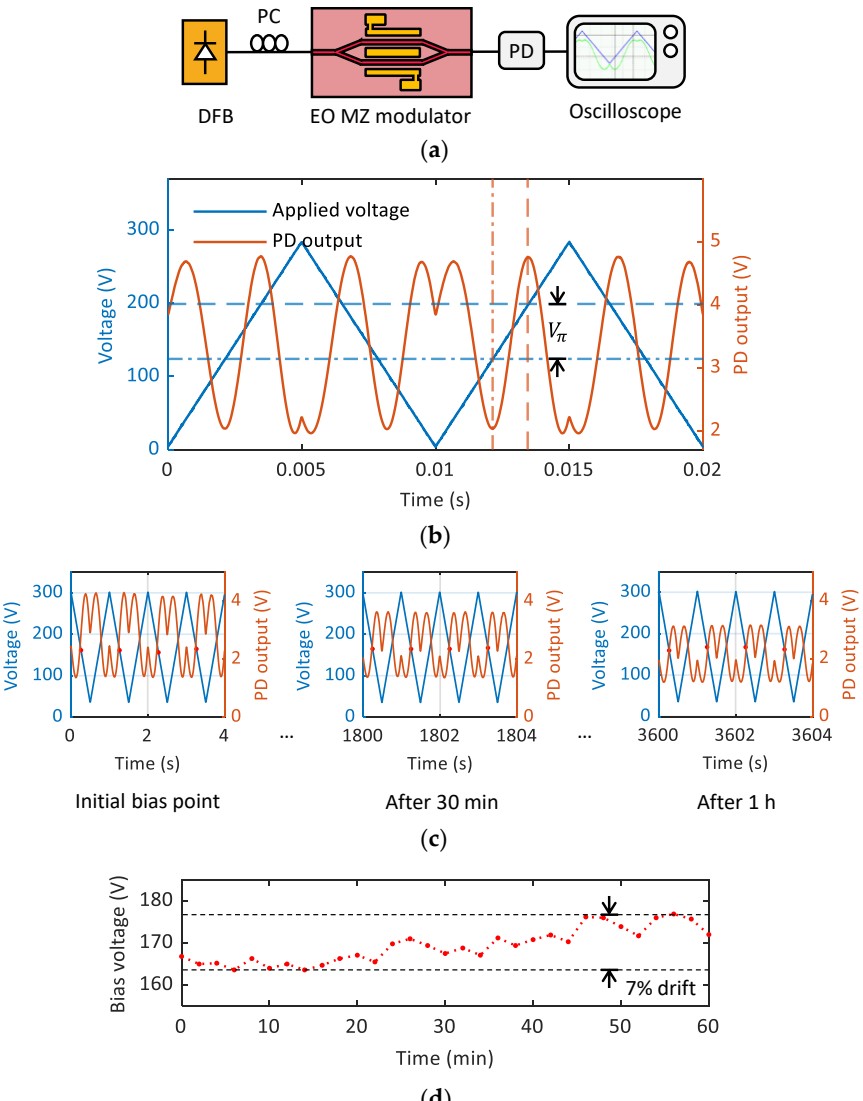

**Figure 8.** Long-term bias-stability experiment results: (**a**) Experimental setup to measure the modulation output signal of the Mach–Zehnder (MZ) modulator and (**b**) intensity modulation signal to find the bias voltage for π–phase difference. (**c**) Output interference signals observed at 0, 0.5, and 1 h, which show no significant change in waveform. (**d**) Bias voltage variation exhibiting 7% of drift occurring in an hour.

The MZ modulator was characterized using the experimental setup illustrated in Figure 8a. The length of the device was 20 mm with an electrode length of 5 mm. The $V_\pi$ of 12 μm gap electrodes on each arm was measured as 75 V, as can be seen in Figure 8b. It was poled with a poling field of 80 V/μm by applying 2.9 kV across the two arms with the middle electrode remaining as floating. The $r_{33}$ calculated from the measured $V_\pi$ was 7.22 pm/V. The bias drift of MZ modulator was observed by monitoring the intensity modulation waveform with a bias voltage of 167 V applied for an hour. As shown in Figure 8c, the waveform changed only slightly in an hour, which indicated that the bias point for quadrature phase difference did not drift significantly. In the case of the ordinary EO polymer device with a vertical field application, the bias voltage was divided proportional to the resistivities of the core and cladding polymers and exhibited a considerably fast drift within a few minutes [4,5]. The bias voltage variation measured for an hour was as small as 7%, as shown in Figure 8d. The amplitude of output waveform was slightly decreased due to the alignment degradation

of butt-coupled fibers. For the longer-term measurement of bias stability, in the next step, we will prepare a complete packaged device to improve the optical alignment stability.

## 5. Conclusions

Based on fluorinated polyimide, a novel EO polymer material was synthesized, and used for the development of an EO polymer modulator with a coplanar electrode, which could provide efficient poling and reduced DC drift. The EO polymer with fluorinated polyimide backbone had the advantages of improved thermal stability and low absorption losses due to the C–H vibration overtone. The poling process was optimized by limiting the maximum current flow during the stepwise voltage increase. An EO coefficient of 8.42 pm/V was obtained from a polarization modulation device at 1550 nm wavelength, which was quite high considering the DR1 chromophore attached to the fluorinated polyimide. The MZ modulator was poled with a lower degree of poling because of the wide electrode gap, and $V_\pi$ was measured to be 75 V with a 5-mm-long electrode, in the push–pull operation. From the measurement of the bias-drift effect, we confirmed that the coplanar electrode could provide significant improvement over the conventional vertical poling device.

To improve the performance of the EO polymer waveguide devices by maximizing the poling efficiency, the device fabrication process must be enhanced so as to reduce the electrode gap. The poling field should be increased, and to do so, the device must be fabricated without any defects causing dielectric breakdown. As the commercial DR1 chromophore has a low EO effect, chromophores with high dipole moments and hyperpolarizabilities should be incorporated as sidechains into the fluorinated polyimides. Then, the EO coefficient can be improved, and EO polymer modulators with low loss, high thermal stability, and low $V_\pi$ can be achieved based on the coplanar electrodes.

**Author Contributions:** Conceptualization, M.-C.O. and E.-S.L.; methodology, E.-S.L.; data analysis, E.-S.L., S.-M.K., and M.-C.O.; material preparation, M.-H.Y. and J.-W.K.; writing—original draft preparation, E.-S.L and M.-C.O.; writing—review and editing, E.-S.L and M.-C.O. All authors have read and agreed to the published version of the manuscript.

**Funding:** This research was supported by National Research Foundation of Korea (NRF) grant funded by the Korea government (MSIP) (2020R1A2C2101562).

**Conflicts of Interest:** The authors declare no conflict of interest.

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
