# Peer review of "Coplanar Electrode Polymer Modulators Incorporating Fluorinated Polyimide Backbone Electro-Optic Polymer"

_photonics, doi:10.3390/photonics7040100_

Round 1

Reviewer 1 Report

The authors have been prepared an incorporating fluorinated polyimide backbone electro-optic polymer, which has the characteristics of high thermal stability and low loss of optical signal. Polarization modulators and Mach Zehnder (MZ) intensity modulators were fabricated by using the material. Based on coplanar electrodes, the device can realize low loss, high thermal stability and low Vπ polymer modulator. The EO coefficients of the devices were measured to be 8.42 pm/V and 7.22 pm/V, and the polarization efficiency was analyzed. This manuscript can be accepted for publication after the following revision.

1. The EO coefficient of polarization modulators and MZI modulators are 7.22 pm/V and 8.42 pm/V respectively. Compared with the other research, the EO coefficient is obviously lower. Why is DR1 chromophore still chosen for its low EO effect?

2. The bias voltage variation measured for an hour was as small as 7%. Why not choose for a long period of time (about 500 hours) to verify the stability of the device?

3. The Bandwidth is an important performance parameter of EO modulator. Can the authors provide or test the normalized frequency response of the modulator?

Author Response

I would like to appreciate the reviewer for the effort to review the manuscript and provide important comments to improve the quality of the manuscript.  In accordance with the reviewer’s comments, I have prepared the responses as described in this letter.  The corrections are also reflected in the revised manuscript.

1st Comment: The EO coefficient of polarization modulators and MZI modulators are 7.22 pm/V and 8.42 pm/V respectively. Compared with the other research, the EO coefficient is obviously lower. Why is DR1 chromophore still chosen for its low EO effect?

Reply: We could not design and synthesize the chromophore itself, and thus we used commercially available chromophore DR1. Though the highly efficient chromophore was not available at this moment, we focused to exhibit the advantage of the coplanar poling.

2nd Comment: The bias voltage variation measured for an hour was as small as 7%. Why not choose for a long period of time (about 500 hours) to verify the stability of the device?

Reply: To perform the longer time measurement, it was necessary to package the sample including the fiber-pigtail. The current setup of optical alignment was not stable enough to maintain the initial bias condition. This comment is reflected in a revised manuscript as “For the longer term measurement of bias stability, in next step, we will prepare a completely packaged device to improve the optical alignment stability.” at 1st paragraph, page 8.

3rd Comment: The Bandwidth is an important performance parameter of EO modulator. Can the authors provide or test the normalized frequency response of the modulator?

Reply: In this device, the driving voltage is too high to modulate at high frequency. The high frequency response will be measured after reducing the Vpi.

Reviewer 2 Report

The manuscript Coplanar Electrode Polymer Modulators Incorporating Fluorinated Polyimide Backbone Electro-Optic Polymer” fits well in several topics covered by the journal Photonics, namely in the fields:

  • Photonic materials and technology
  • Integrated Optoelectronics and Integrated Optics
  • Photonics Device and Technologies
  • Fundamental & applications of photonics / optics

The manuscript covers important issues of the design, fabrication, and characterization of  electro-optic polymer-based modulators, which are fundamental devices in optical transmission systems. The manuscript is in general well-structured and the results clearly presented.

The presented work represents an original work with scientific content, where the methods and results were adequate. The demonstrated EO devices produced with an innovative material could provide increased poling efficiency and reduced DC drift, being therefore a potential alternative to mainstream EO polymer-based modulators.

From the stated above and apart from the suggested corrections (Manuscript remarks for Revision), I propose the publication of the manuscript with minor revisions.

Author Response

I would like to appreciate the reviewer for the effort to review the manuscript and provide important comments to improve the quality of the manuscript.  In accordance with the reviewer’s comments, I have prepared the responses as described in the attached letter.  The corrections are also reflected in the revised manuscript. Please see the attachment.

Reviewer 3 Report

Min-Cheol Oh and collaborators report on the development of an electro-optic polymer modulators with improved thermal stability. I think that the manuscript describes a very good incremental work, which can be of interest for the community. Therefore, I recommend publication after having revised the following minor issue:

Page 1, lines 29-38: The authors should insert appropriate references related to the development of polymers that are resilient against photo-oxidation, with reduced bias drift and insertion losses.

Author Response

Response to the Comments from the 3rd Reviewer

I would like to appreciate the reviewer for the effort to review the manuscript and provide important comments to improve the quality of the manuscript.  In accordance with the reviewer’s comments, I have prepared the responses as described in this letter.  The corrections are also reflected in the revised manuscript.

1st comment: Page 1, lines 29-38: The authors should insert appropriate references related to the development of polymers that are resilient against photo-oxidation, with reduced bias drift and insertion losses.

Reply: Organic materials must have the photo-oxidation issue unless it is hermetically packaged. It is deeply investigated in the field of OLED researches. However, there are not many experimental investigations regarding the photo-oxidation of the EO polymers except reference [4]. Moreover, EO polymer devices have not yet been able to prove practically useful in commercial applications requiring perfect performance without the problems of photo-oxidation, bias drift, and low insertion loss.